# Modulating myoblast differentiation with RNA-based controllers

**Peter B. Dykstra**[1], **Thomas A. Rando**[2,3¤], **Christina D. Smolke**[1,4]*

**1** Department of Bioengineering, Stanford University, Stanford, CA, United States of America, **2** Department of Neurology and Neurological Sciences, Stanford University School of Medicine, Stanford, CA, United States of America, **3** Paul F. Glenn Center for the Biology of Aging, Stanford University School of Medicine, Stanford, CA, United States of America, **4** Chan Zuckerberg Biohub, San Francisco, CA, United States of America

¤ Current address: Broad Stem Cell Research Center, University of California Los Angeles, Los Angeles, CA, United States of America

* csmolke@stanford.edu

**Data Availability Statement:** All relevant data are within the manuscript and the Supporting Information files.

**Funding:** This work was supported by grants from the National Institutes of Health to C.D.S. (R01 GM086663) and to T.A.R. (R01 AG068667, R01

## Abstract

Tunable genetic controllers play a critical role in the engineering of biological systems that respond to environmental and cellular signals. RNA devices, a class of engineered RNA-based controllers, enable tunable gene expression control of target genes in response to molecular effectors. RNA devices have been demonstrated in a number of systems showing proof-of-concept of applying ligand-responsive control over therapeutic activities, including regulation of cell fate decisions such as T cell proliferation and apoptosis. Here, we describe the application of a theophylline-responsive RNA device in a muscle progenitor cell system to control myogenic differentiation. Ribozyme-based RNA switches responsive to theophylline control fluorescent reporter expression in C2C12 myoblasts in a ligand dependent manner. HRAS and JAK1, both anti-differentiation proteins, were incorporated into RNA devices. Finally, we demonstrate that the regulation of HRAS expression via theophylline-responsive RNA devices results in the modulation of myoblast differentiation in a theophylline-dependent manner. Our work highlights the potential for RNA devices to exert drug-responsive, tunable control over cell fate decisions with applications in stem cell therapy and basic stem cell biology research.

## Introduction

Synthetic biology offers a framework to design, construct, and reprogram biological systems. Genetic elements such as promoters, genes, and regulatory modules can be engineered to respond to various inputs and produce a desired output. These elements can be modified to specific purposes depending on the application. Canonical examples of early synthetic biology systems include toggle switches [1] and oscillators [2], but more recent efforts have highlighted the applications of genetic circuits in mammalian cells [3–6]. RNA engineering has emerged as an exciting subfield of synthetic biology, as RNA offers customizable ligand-sensing and gene-regulatory elements with tunable response characteristics [7]. RNA-based regulatory systems

AR073248, and P01 AG036695) and from the
National Science Foundation (graduate fellowship
to P.B.D.). C.D.S. is a Chan Zuckerberg Biohub
investigator. The funders had no role in study
design, data collection and analysis, decision to
publish, or preparation of the manuscript.

**Competing interests:** The authors have declared
that no competing interests exist.

have been demonstrated to provide ligand-responsive control over gene expression in mammalian cells and to modulate cellular processes central to cell fate decisions [8, 9]. These ligand-responsive RNA control devices have been developed by coupling RNA aptamers and RNA gene-regulatory elements in a manner that allows ligand binding to the RNA aptamer to modulate the activity of the gene-regulatory element, thus resulting in ligand-responsive gene expression control [10]. Recent examples include inducible control of gene therapy [11] and control of gene editing and transcriptional regulation with CRISPR [12].

One class of RNA control devices utilizes a self-cleaving ribozyme as the gene-regulatory element. The ribozyme-based switch can be placed in the 3' untranslated region (UTR) of a target transcript, where ribozyme cleavage results in rapid degradation of the mRNA and thus lowered gene expression. Ligand binding to an aptamer sequence in the device results in a ribozyme-inactive conformation, causing increased gene expression and thereby acting as a gene expression ON-switch. These RNA-based regulatory systems have been shown to exhibit ligand-responsive control over mammalian gene expression and have modulated cell processes such as proliferation [8].

Cell differentiation and tissue engineering offer a promising application space for synthetic biology approaches [13]. Regenerative medicine and tissue engineering rely on the ability of stem cells to proliferate, self-renew, and differentiate. However, tools to guide these cell fate decisions typically rely on external modalities such as static materials and signals with off-target or immunogenic effects. Skeletal muscle is of particular interest to the tissue engineering field due to its remarkable regenerative capacity, which is controlled by muscle stem cells (MuSCs) [14]. During the regenerative process *in vivo*, or in response to serum-deficient medium *in vitro*, MuSCs activate and undergo differentiation and the resulting myoblasts fuse together to form multinucleated myotubes [14–16]. Current cell-based therapies for engineering muscle tissue face several challenges. Treatments are hampered by a limited number of cells available to transplant due to loss of potency during *in vitro* culture and constraints on the transplanted volume. Therapeutic strategies based on MuSC transplantation also suffer from limited expansion capability *in vivo* as cells tend to differentiate and engraft quickly *in vivo* [17]. Mitigating the push towards differentiation and allowing a cell population to continue expanding *in vivo* is one path forward.

Here we demonstrate a theophylline-responsive RNA-based regulatory system that provides conditional modulation of muscle myofiber differentiation in C2C12 myoblasts. Fluorescent reporter protein expression responsive to theophylline showcases the application of RNA switches in a novel muscle cell type. We highlight RNA devices applied to the conditional regulation of HRAS and JAK1 proteins that affect cell proliferation and differentiation. Finally, we show the efficacy of HRAS-based RNA controllers by demonstrating theophylline-responsive inhibition of myoblast differentiation as well as theophylline-responsive inhibition of myotube fusion. Such a platform shows the potential for future devices to exert engineered drug-responsive, tunable control over cell fate decisions with applications in stem cell therapy as well as basic stem cell biology research.

## Results

### A ribozyme-based regulatory system

RNA devices are engineered by incorporating distinct sensor and actuator components into a coherent regulatory system. Sensor and actuator components are typically connected so that a ligand binding to the sensor affects actuator activity. In previous work, self-cleaving ribozymes were modified to respond to specific small molecules by designing ligand-binding aptamer sequences into the ribozyme [18]. Typically, the ribozyme switch containing both the sensor

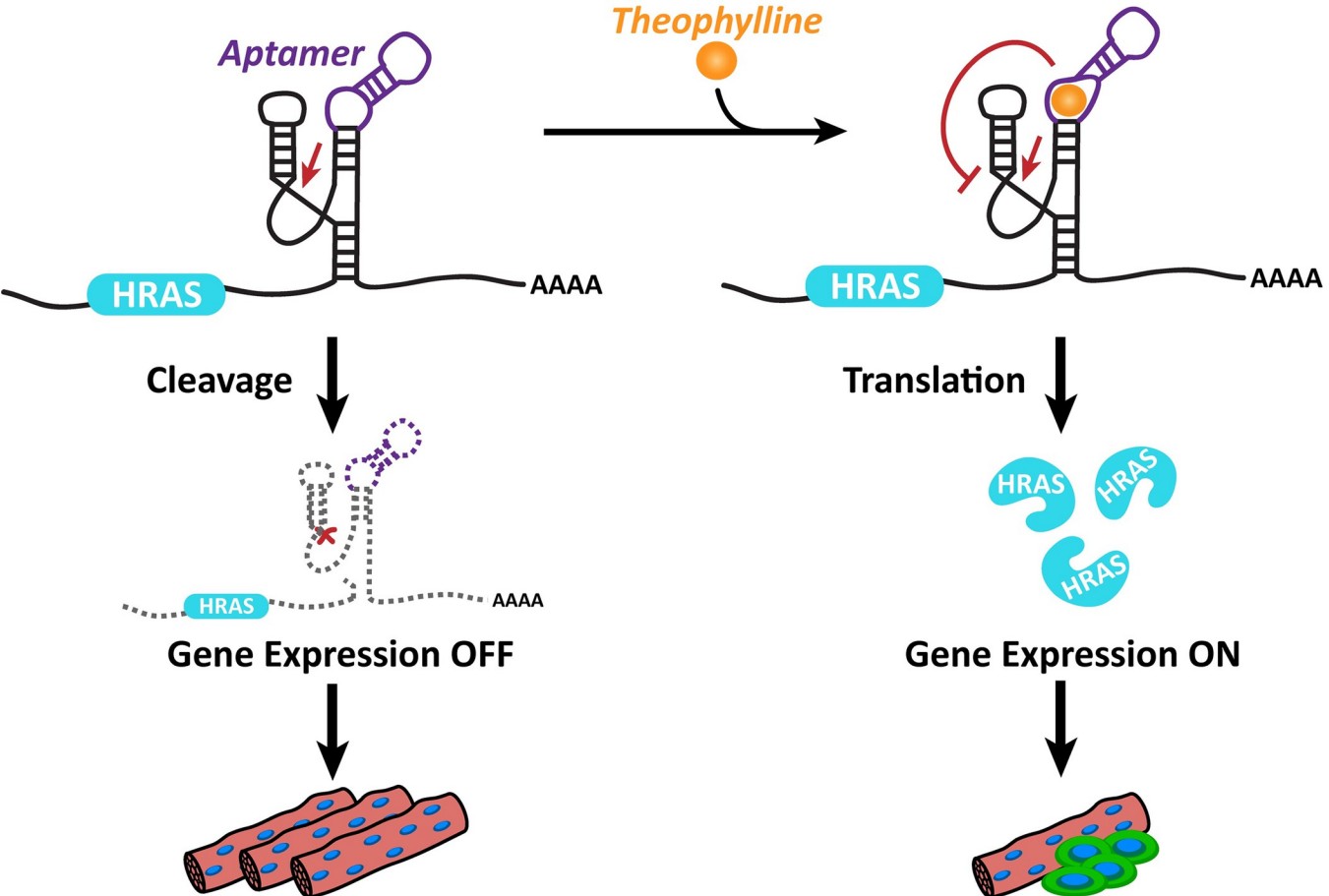

**Fig 1. A ribozyme-based regulatory system.** A ribozyme is inserted into the 3' untranslated region between the translation termination codon of the gene coding sequence and the poly-A tail. The ribozyme contains an aptamer to theophylline (purple) integrated into one of the hammerhead ribozyme's two loops. In the absence of theophylline, the ribozyme cleaves at the indicated point (red arrow). Upon cleavage, the gene coding sequence is separated from the poly-A tail and the transcript is degraded, turning gene expression OFF. When theophylline is present, the loop-based aptamer binds the theophylline molecule and the resulting conformation change prevents the ribozyme from self-cleaving. The coding sequence is subsequently translated into protein and gene expression is ON.

and the actuator is located in the 3' UTR of a transcript coding for a protein of interest (Fig 1). Upon ribozyme self-cleavage, the mRNA transcript is degraded and gene expression is decreased. When a ligand binds the aptamer sequence, the ribozyme structure changes to a ribozyme-inactive confirmation and gene expression is increased. Thus, the sensor and actuator components function together with a gene of interest as a gene expression ON-switch.

### Theophylline-responsive ribozyme switches demonstrate activity in myoblasts

The first step in building a genetic control system for modulating myoblast differentiation was to identify ligand-responsive ribozyme switches that exhibit sufficient activity in the cell line of interest. We designed and tested an RNA-based regulatory system that had previously been characterized in *Saccharomyces cerevisiae*, HEK cells, and mammalian T cells [8, 19, 20]. These devices incorporate an aptamer sequence to sense biomolecules and a self-cleaving ribozyme as an actuating module to provide gene regulation. RNA aptamer sequences to theophylline have provided the greatest activation ratio and thus dynamic range in prior studies [20]. In

particular, two theophylline-responsive ribozyme switch sequences previously described, with loop II sequences CAUAA and AGAAA, demonstrated high activation ratios when characterized in HEK293T cells [20].

A model cell system for studying muscle was used to characterize genetic designs in the skeletal muscle context. The C2C12 cell line is a mouse myoblast line that overcomes inherent limitations of primary cells, such as small cell numbers and limited expansion capacity [21]. C2C12 cells are robust and capable of rapid proliferation in high-serum conditions. Under low-serum conditions, however, these cells exhibit a dramatic differentiation into the myogenic program in which mononucleated myoblasts fuse to form multinucleated myotubes and subsequent fully-fledged contractile skeletal muscle cells [22].

Ribozyme switch characterization constructs were designed and tested in C2C12 myoblasts as switch activity can differ between cell types. The ribozyme switch characterization cassettes include an EF1α promoter driving expression of mCherry fluorescent protein under the control of a theophylline-responsive switch as well as a PGK promoter driving expression of Clover fluorescent protein to act as a transfection control (Fig 2A). Two ribozyme switches with loop II sequences CAUAA (TheoCAUAA) and AGAAA (TheoAGAAA) were tested alongside a wild-type satellite RNA of the Tobacco Ringspot Virus (sTRSV) hammerhead ribozyme representing maximal cleavage and a non-cleaving mutated sTRSV ribozyme (sTRSVctrl) representing minimal cleavage and maximal expression. Constructs were transiently transfected into C2C12 cells and cells were grown for two days in growth medium containing 0 or 1 mM theophylline. Comparison of the gene expression activity of four switches demonstrated a clear theophylline-responsiveness of the switches, with activation ratios of 5.7 for TheoCAUAA and 6.0 for TheoAGAAA (Fig 2B). The observed dynamic range was slightly less than that observed for these same switches in HEK293T cells [20]. The TheoAGAAA switch was further characterized to obtain a dose-response curve connecting fluorescence levels to theophylline concentration, with an $EC_{50}$ of 269 μM (Fig 2C). Notably, while some cell toxicity was observed at theophylline concentrations of 1 mM or higher (S1 Fig), ribozyme switches in viable cells maintained high levels of fluorescent protein expression. Taken together, these results show that theophylline-responsive ribozyme switches function effectively in C2C12 myoblasts.

## Anti-differentiation signals enable a myogenic regulatory system

The next step in building a genetic control system for modulating myoblast differentiation was to identify proteins known to affect differentiation of myoblasts. Strong anti-differentiation signals can take the form of strong pro-proliferation signals. One such type of proliferation signal can be found in a class of genes called oncogenes that have the potential to cause cells to become cancerous when mutated. One such oncogene is HRAS which codes for a GTPase called H-Ras that plays an important role in regulating cell division [23]. We selected HRAS for use in this study because it offers an exceptionally strong anti-differentiation signal without necessarily being related to the typical myogenic program. The version of HRAS used in this work contains a mutation replacing glycine with valine at position 12 (G12V) causing the protein to remain permanently activated leading to high levels of proliferation. This HRAS protein provides a strong pro-proliferation signal through the MAPK/ERK pathway [23] (Fig 3A). In contrast, Janus kinase 1 (JAK1) is an anti-differentiation signal that has been identified as part of an important pathway promoting myoblast proliferation (Fig 3A) [24]. The JAK signal transducer and activator of transcription (STAT) pathway is well-characterized [25]. Overexpression of JAK1 has been shown to lower expression of Myogenin and Myosin Heavy Chain (MyHC) [24], two key markers of myogenic differentiation [26, 27].

**A.**

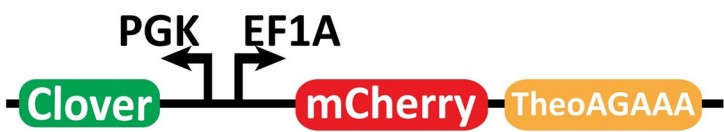

**B.** **C.**

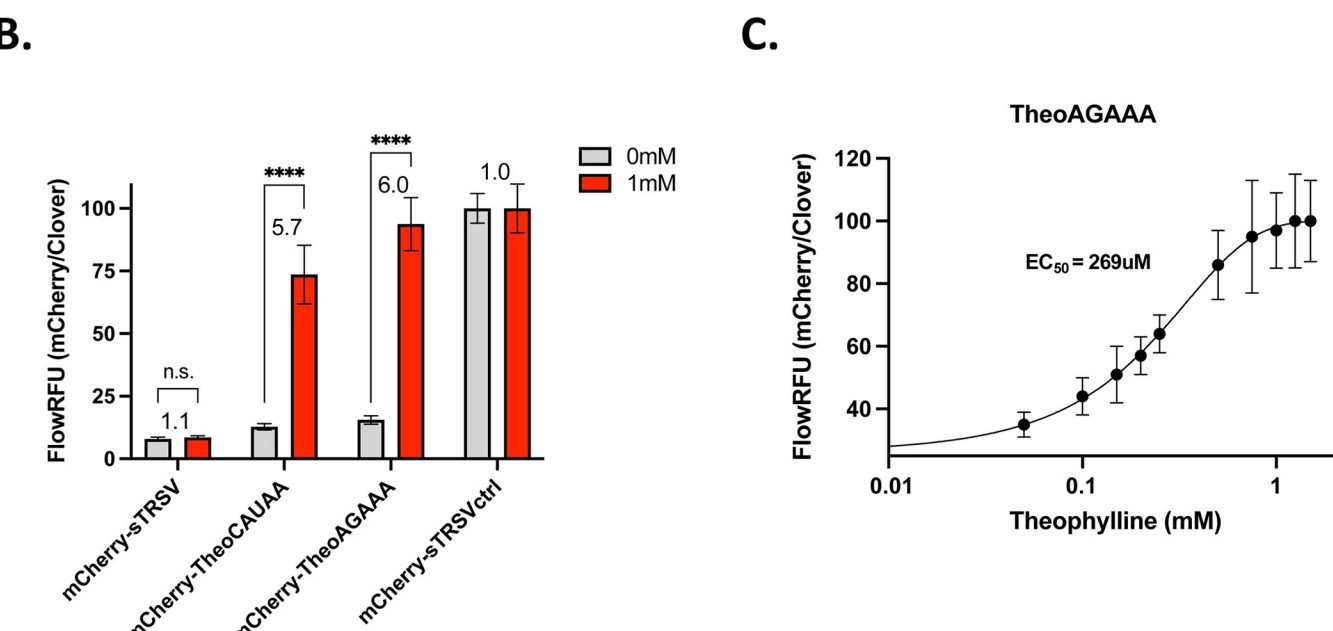

**Fig 2. Theophylline-responsive ribozyme switches demonstrate activity in myoblasts. A.** Diagram of a bicistronic fluorescent reporter construct. mCherry expression driven by an EF1α promoter is controlled by a theophylline-responsive switch. Clover expression driven by a PGK promoter offers a transfection control for normalization. **B.** Flow cytometry analysis of multiple theophylline-responsive ribozyme switches. For each sample, the mean fluorescence intensity of mCherry was normalized by the mean fluorescence intensity of Clover to give the mean fluorescence ratio. The mean fluorescence ratio was normalized to that of the non-cleaving sTRSVctrl and scaled by a factor of 100 to give relative fluorescence values (FlowRFU). Activation ratio (AR) of mCherry/Clover representing fold change of normalized mCherry expression is indicated above each set of bars. sTRSV is the wild-type hammerhead ribozyme; sTRSVctrl is a non-cleaving mutant of sTRSV; error bars indicate standard deviation of four biological replicates; asterisks indicate significance based on p-values from t-test, n.s. indicates not significant, * $p < 0.01$, ** $p < 0.001$, *** $p < 0.0001$, **** $p < 0.00001$. **C.** Dose response curve for the TheoAGAAA ribozyme switch showing relative fluorescence values as a function of theophylline concentration. $EC_{50}$ value reported as mean of four biological replicates; error bars indicate standard deviation of four biological replicates.

Initial characterization of HRAS and JAK1 as anti-differentiation signals in the C2C12 model cell line was performed by designing constructs with either HRAS or JAK1 linked to the sTRSVctrl non-cleaving ribozyme control for achieving the maximal expression state. C2C12 cells were transiently transfected with the HRAS- and JAK1-overexpression constructs and grown in growth medium for two days until the cells were 80–90% confluent. Cells were then switched to differentiation medium for three days. Cells were then imaged to assess differentiation extent. The ratio of cell nuclei associated with MyHC+ myofibers relative to the total number of cell nuclei (i.e., the fusion index) was determined for each group. HRAS-sTRSVctrl expression demonstrated a 64% decrease in the fusion index compared to wild-type negative controls (Fig 3B and 3C). The number of total cell nuclei observed increased 129%, consistent with a promotion of proliferation by HRAS in addition to the inhibition of differentiation (Fig

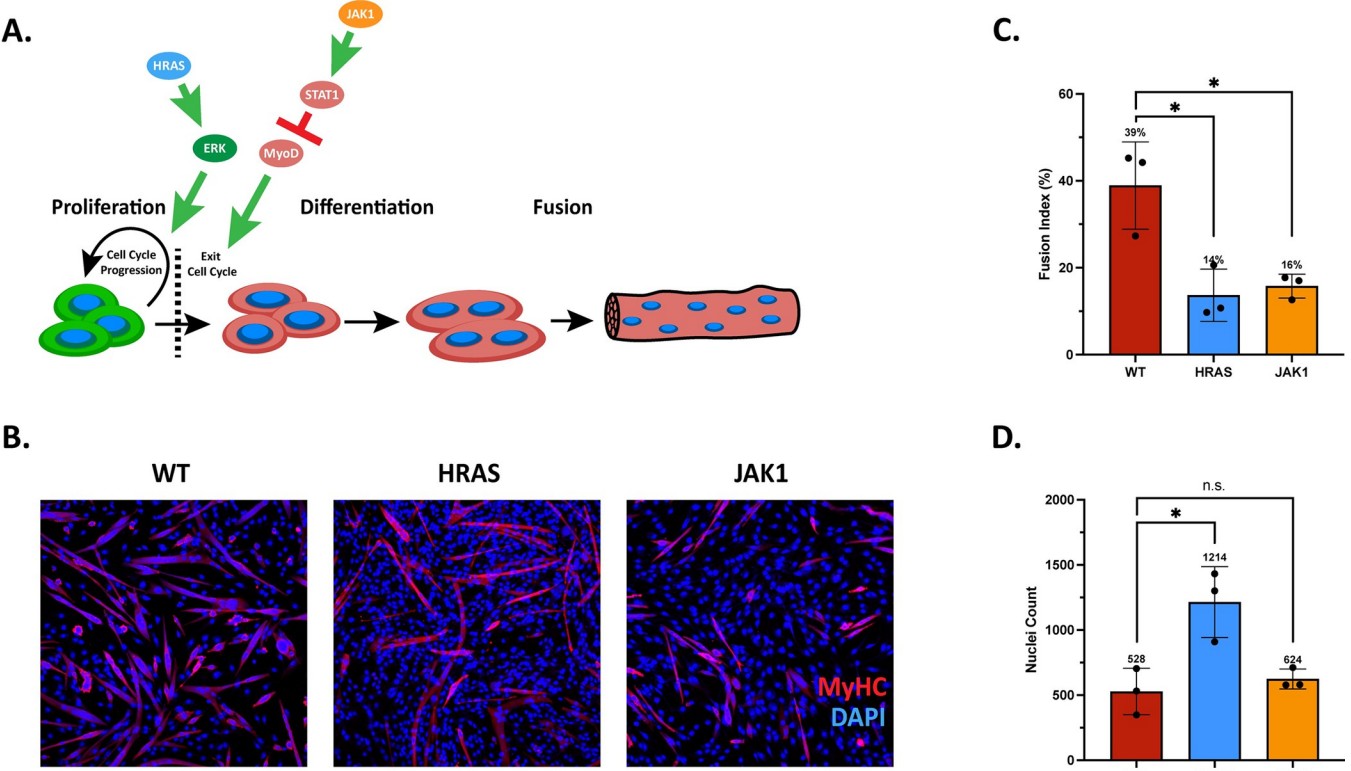

**Fig 3. Anti-differentiation signals enable a myogenic regulatory system. A.** Diagram of myoblast proliferation and differentiation. Myoblasts in a proliferative state will exit the cell cycle in response to low-serum medium. Myoblasts committed to differentiation express Myogenin before fusing together to form multinucleated myotubes that express MyHC. HRAS promotes proliferation via the MAPK/ERK pathway, causing continued cell growth. JAK1 promotes STAT1 expression, which inhibits MyoD, a key mediator of muscle differentiation. **B.** Representative immunofluorescence microscopy images of wild type (WT) and HRAS- and JAK1-overexpressed C2C12 cells. MyHC expression is shown in red while DAPI, a dye that stains DNA, is shown in blue. **C.** Quantification of fusion index from WT and HRAS- and JAK1-overexpressed C2C12 cells. **D.** Quantification of total nuclei from WT and HRAS- and JAK1-overexpressed C2C12 cells. Values are means; error bars indicate standard deviation of three biological replicates; asterisks indicate significance based on p-values from t-test, n.s. indicates not significant, * p < 0.01, ** p < 0.001, *** p < 0.0001, **** p < 0.00001.

3D). Cells expressing JAK1-sTRSVctrl exhibited a 59% decrease in differentiation compared to wild-type negative controls (Fig 3B and 3C), while the total nuclei count remained similar to wild-type (Fig 3D). Comparison of non-cleaving sTRSVctrl constructs to constitutively cleaving sTRSV constructs reveals a decrease in markers of differentiation and highlights the usefulness of sTRSVctrl and sTRSV as control constructs (S2 Fig). Taken together, the data indicate that overexpression of HRAS and JAK1 generate anti-differentiation effects.

## HRAS-based RNA controllers exhibit theophylline-dependent inhibition of myoblast differentiation

In order to build a ligand-responsive RNA device to regulate myoblast differentiation, we next coupled the HRAS and JAK1 proteins with the theophylline-responsive ribozyme switches, TheoCAUAA and TheoAGAAA, that were validated with a fluorescent reporter in C2C12 cells. Characterizing the RNA controller constructs required detecting HRAS and JAK1 transcripts to measure the direct output of the RNA device at different theophylline concentrations. We thus examined whether the RNA controllers can regulate HRAS and JAK1 expression in a ligand-responsive manner. HRAS or JAK1 sequences were incorporated into constructs with sequences coding for the ribozyme switches, TheoCAUAA and TheoAGAAA,

and two control constructs, the wild-type sTRSV and non-cleaving sTRSVctrl. Plasmids encoding the RNA controller expression cassettes were transiently transfected into C2C12 cells which were grown and differentiated in media containing 0 or 1 mM theophylline as described previously. Transcript levels of HRAS and JAK1 were probed via RT-qPCR one day (T1) after transient transfection. Transcript expression at T1 was normalized to that from sTRSVctrl (or the maximum transcript expression) to eliminate any side effects of theophylline and is represented as the fold change from sTRSVctrl. HRAS expression at T1 exhibited 2.9 and 3.1-fold activation ratios for the HRAS-TheoCAUAA and HRAS-TheoAGAAA constructs in the presence of 1 mM theophylline (Fig 4A). Similarly, JAK1-TheoCAUAA and JAK1-TheoAGAAA constructs exhibited activation ratios of 3.2-fold and 3.8-fold, respectively (Fig 4B). TheoCAUAA and TheoAGAAA were also shown to control transcript levels across a range of doses between 0 mM and 1 mM theophylline (S3 Fig). Taken together, these results demonstrate that the theophylline-responsive ribozyme switches control the expression of the HRAS and JAK1 transcripts in a theophylline-responsive manner.

We next examined whether regulation of HRAS and JAK1 expression in a ligand-responsive manner via the RNA controllers led to downstream changes in myoblast differentiation. Myogenin and MyHC were selected as markers of myoblast differentiation due to their well-characterized importance in the myogenic program [28]. Transcript levels for both differentiation markers were examined at three days after the addition of differentiation medium (i.e., five days post-transfection, T5) to allow time for differentiation to take place. Transcript data at T5 was normalized to that achieved from sTRSV (or the maximum expression of differentiation markers), as the highest levels of differentiation markers corresponded with the lowest expression of the anti-differentiation protein (HRAS or JAK1). The data are presented as the fold change from sTRSV. Substantial inhibition of differentiation was observed in the expression of Myogenin from both the HRAS-TheoCAUAA (42% decrease) and HRAS-TheoA-GAAA (27% decrease) constructs in the presence of 1 mM theophylline (Fig 4C). This theophylline-dependent effect was also observed to a greater degree in the expression of MyHC. MyHC expression was decreased up to 49% in the presence of 1 mM theophylline for the HRAS-TheoCAUAA construct and decreased up to 35% for HRAS TheoAGAAA (Fig 4D). The JAK1-based RNA controller constructs did not demonstrate the same anti-differentiation effect in either the expression of Myogenin (Fig 4E) or the expression of MyHC (Fig 4F). Notably, there was no significant effect on Myogenin or MyHC observed in the JAK1-sTRSVctrl constructs compared to the JAK1-sTRSV constructs. Taken together, the results demonstrate the efficacy of HRAS-based RNA controllers on transcript-level markers of differentiation. Furthermore, the negative results seen with JAK1-based RNA controllers highlight the relative difference in pro-proliferation power of HRAS and JAK1 signals in this context.

## HRAS-based RNA controllers demonstrate theophylline-responsive inhibition of myoblast fusion

Examination of phenotypic changes in differentiation offers an additional layer of analysis how the RNA controller system can play a role in modulating myoblast differentiation. Fusion index, as previously described, serves as a visible and quantifiable metric to evaluate myogenic differentiation. Due to the negative results from the JAK1-based RNA controllers tested previously, only HRAS-based RNA controllers were used in the fusion index assay. C2C12 cells were transiently transfected with HRAS-based RNA controllers and maintained in growth and differentiation media over the course of five days until cells were fixed and labeled at T5. Representative images of differentiating myoblasts at T5 show comparisons between cell

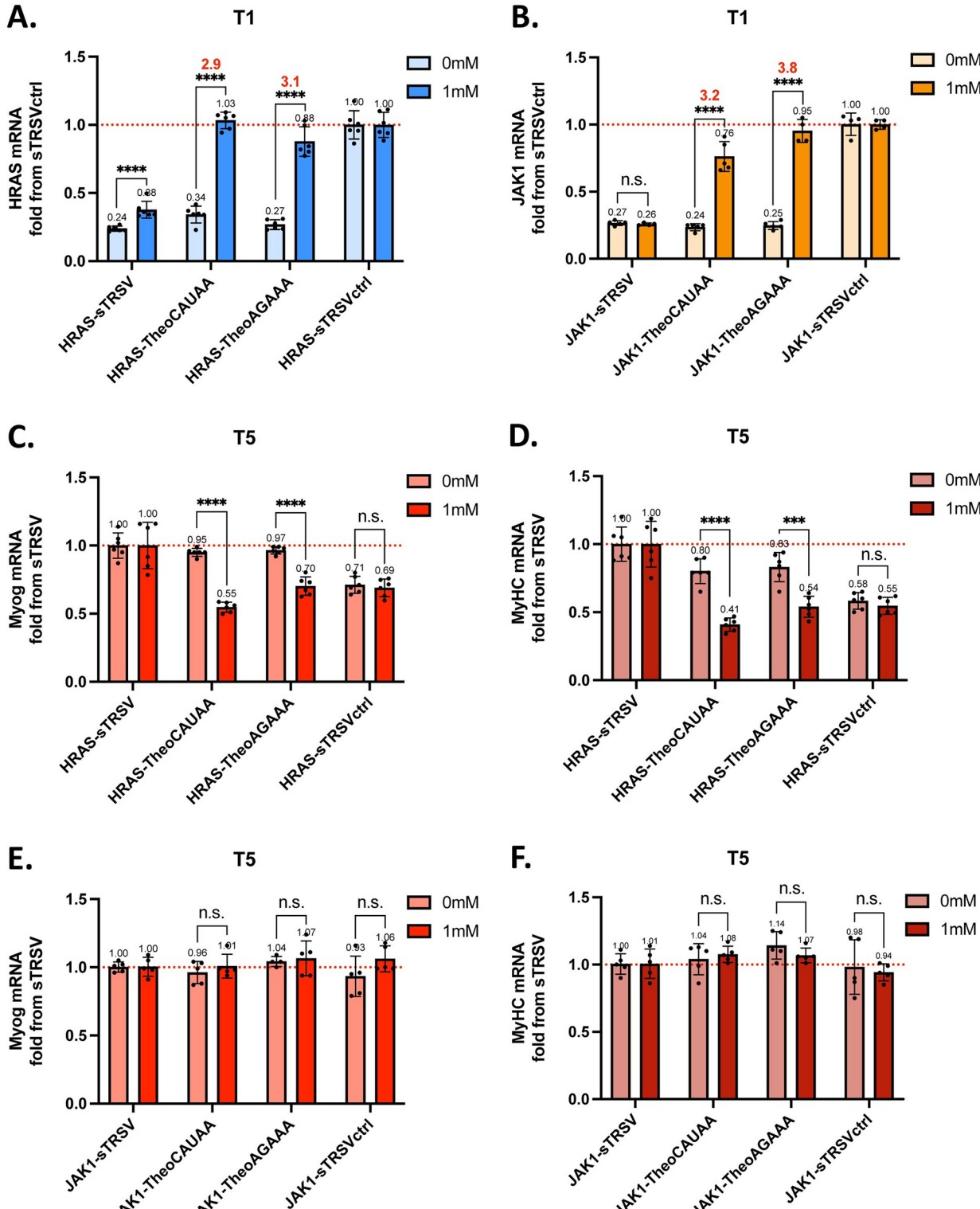

**Fig 4. HRAS-based RNA controllers exhibit theophylline-dependent inhibition of myoblast differentiation. A.** Real-time RT-qPCR detection of HRAS expression one day after transient transfection of HRAS constructs. **B.** Real-time RT-qPCR detection of JAK1 expression one day after transient transfection of JAK1 constructs. **C.** Real-time RT-qPCR detection of Myogenin five days after transient transfection of HRAS constructs. **D.** Real-time RT-qPCR detection of MyHC five days after transient transfection of HRAS constructs. **E.** Real-time RT-qPCR detection of Myogenin five days after transient transfection of JAK1 constructs. **F.** Real-time RT-qPCR detection of MyHC five days after

transient transfection of JAK1 constructs. Activation ratio (AR) representing fold change of the indicated transcript is indicated above each set of bars. sTRSV is the wild-type hammerhead ribozyme; sTRSVctrl is a non-cleaving mutant of sTRSV; Values are means normalized to either sTRSVctrl (A and B) or sTRSV (C-F); error bars indicate standard deviation of four or more biological replicates; asterisks indicate significance based on p-values from t-test, n.s. indicates not significant, $^*$ $p < 0.01$, $^{**}$ $p < 0.001$, $^{***}$ $p < 0.0001$, $^{****}$ $p < 0.00001$.

populations expressing various HRAS-based RNA controllers (Fig 5A). Notably, the number of cell nuclei visible when C2C12 cells express the HRAS-sTRSVctrl construct show the pro-proliferation capacity of HRAS in this context. Also, additional evidence of a side effect of the-ophylline could be observed in the decreased number of cells in all samples with media con-taining 1 mM theophylline (bottom row). Quantification of the fusion index demonstrated theophylline-dependent inhibition of myoblast fusion from both HRAS-TheoCAUAA and HRAS-TheoAGAAA constructs. Fusion index data at T5 was normalized to that achieved from sTRSV as the maximum amount of differentiation corresponded with the lowest expres-sion of HRAS via the HRAS-sTRSV construct. Data are presented as the fold change from sTRSV. The fusion index was decreased by 47% in the presence of 1 mM theophylline for the HRAS-TheoCAUAA construct and by 30% for the HRAS-TheoAGAAA construct (Fig 5B). The magnitude of these reductions in differentiation measured by fusion index mirrored those found via RT-qPCR. Taken together, the results provide further support that HRAS-based RNA controllers can modulate myoblast differentiation in a theophylline-dependent manner.

## Discussion

In the scope of this study, we applied ligand-responsive RNA devices to a novel cell type and cell fate process. We validated activities of theophylline-responsive ribozyme switches in C2C12 myoblasts using fluorescent reporters and incorporated anti-differentiation proteins. We designed and tested HRAS-based and JAK1-based RNA controllers in C2C12 myoblasts and demonstrated inhibition of myoblast differentiation in a theophylline-dependent manner with an HRAS-based ribozyme device. Transcript levels of MyHC were reduced up to 49% with HRAS-TheoCAUAA. We further showed theophylline-dependent inhibition of myoblast fusion with HRAS-based controllers, with up to a 47% reduction in fusion index with HRAS-TheoCAUAA. Taken together, these results demonstrate a novel application of ligand-responsive RNA devices and highlight the potential of future devices for tunable control over cell fate decisions.

Earlier applications of ribozyme-based RNA devices focused on controlling the expression of cytokines in T cell proliferation by upregulating pro-proliferative cytokines in the presence of a ligand [8, 9]. This ligand-responsive approach enables tunable control inherent in the RNA device, an advantage for future translational strategies. In these T cell applications, the default state in the absence of ligand is suppression of cell growth and the addition of ligand to the system turns on a self-reinforcing pro-proliferation signal. Myoblast differentiation offers a different challenge. Cells are induced to stop dividing and proceed through differentiation when cultured in low-serum differentiation medium. Because the cells experience a strong extrinsic push toward differentiation from the low-serum medium, observing large effect sizes from the upregulation of pro-proliferation and anti-differentiation signals is challenging. The differences observed in our results between the effects of the HRAS- and JAK1-based control constructs highlight this dilemma. The oncogene HRAS is an incredibly strong signal, yet it managed to slow differentiation by only a moderate amount when examining transcript levels of Myogenin and MyHC. In this same context, JAK1 proved not a strong enough signal to see any significant effect on differentiation. One limitation of the experimental design is that the maximal transgene expression from the genetic constructs might not have been sufficient to

## A.

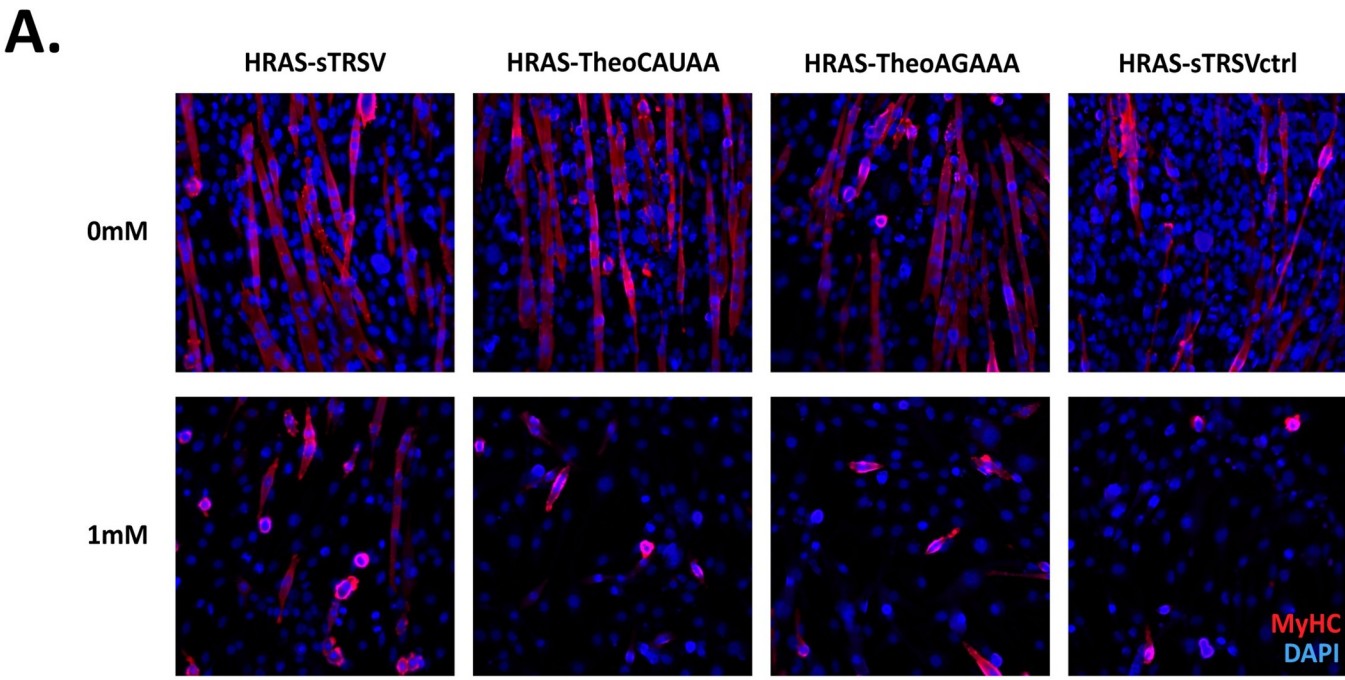

## B.

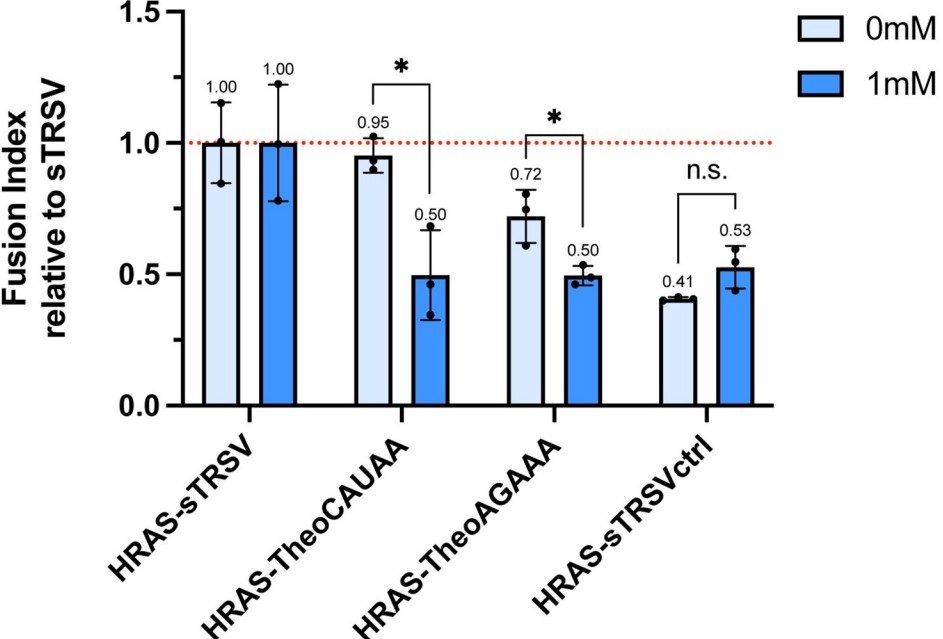

**Fig 5. HRAS-based RNA controllers demonstrate theophylline-responsive inhibition of myoblast fusion. A.** Representative immunofluorescence microscopy images at T5 of C2C12 cells transfected with the indicated HRAS constructs and cultured in medium with 0 or 1 mM theophylline. MyHC expression is shown in red while DAPI is shown in blue. **B.** Quantification of fusion index with HRAS-based RNA controller constructs. Fusion index values are means normalized to sTRSV; error bars indicate standard deviation of three biological replicates; asterisks indicate significance based on p-values from t-test, n.s. indicates not significant, * p < 0.01, ** p < 0.001, *** p < 0.0001, **** p < 0.00001.

cause a larger response. The transient effect of transfecting the RNA controller constructs could have reduced the longer-term effectiveness of protein expression and instead relied on a bolus of pro-proliferative signal early in the differentiation process to carry an effect through the time course of the experiment. Furthermore, small-amplitude regulation of JAK1 expression could result in concurrent up- or down-regulation in other redundant genes to preserve the myogenic pathway. An alternative strategy to approach the process of differentiation might be to stably transfect or transduce the RNA controller construct in order to produce a longer-term and more sustained effect. Cells with an integrated RNA controller could also be generated via CRISPR/Cas systems. In addition, other proteins might be identified that inhabit more relevant nodes in the myogenic pathway that benefit from positive feed-forward loops and signaling cascades to maximize downstream effects.

Another consideration for the design of these types of RNA controllers is the choice of ligand and the potential of unexpected side effects. Theophylline is associated with cell toxicity at higher concentrations [29], however, it is possible that low to moderate concentrations of theophylline have an effect on cell growth that resulted in downstream effects on cell differentiation. To account for these effects in this study, data was normalized within the 0 and 1 mM conditions to address any direct effect of theophylline on differentiation and isolate the theophylline-dependent device activity. However, attempts at building upon this work could also incorporate aptamers to ligands other than theophylline. Swapping out theophylline for a different ligand could eliminate any toxicity effects on the cell fate process of interest. Aptamers to hypoxanthine, cyclic-di-GMP, and folinic acid have already been incorporated into RNA devices and could be examined in this context [20].

Future work may focus on building RNA controllers with alternative proteins or regulatory mechanisms for the control of myoblast differentiation. Identifying a suitable alternative to HRAS that utilizes native cell signaling pathways around proliferation and differentiation to effectuate a strong anti-differentiation signal would be ideal. For example, ALKBH1, a $N^6$-methyladenine demethylase, and TGFBR1, a receptor in the TGF-β/Smad signaling pathway, have both been shown to enhance proliferation and inhibit differentiation upon overexpression in C2C12 cells [30, 31]. Future studies could also examine targets early in the myogenic pathway, i.e., Myf5 or MyoD [15], as well as an array of genes directly associated with myotube fusion, including myomaker and myomerger [32]. Regardless of the target, while gene expression changes resulting from the RNA controllers were confirmed by a transcript-based qRT-PCR assay in this study, they could also be confirmed with a protein-based assay such as a Western blot to analyze expression of the target protein. Another strategy could be to modify the design of the RNA control system. For example, miRNA-based controllers have been demonstrated to act as OFF-switches targeting endogenous protein expression [9]. This approach could be pursued as a stand-alone method or in conjunction with one of the ON-switches presented in this work. Such a miRNA platform could incorporate miRNAs shown to regulate myoblast proliferation and differentiation, such as miR-133, miR-452, and miR-664-5p [33–35].

Overall, this work provides a proof-of-concept highlighting the application of theophylline-responsive RNA controllers in myoblast differentiation. Advances in RNA aptamer discovery and design will expand the number of small-molecule or protein inputs for a range of genetic circuit designs. Testing other alternative proteins could provide an anti-differentiation signal with closer ties to the canonical myogenic signaling pathway. In addition, alternative RNA device architectures, for example utilizing miRNA switches, would allow orthogonal targeting of endogenous genes to drive cell fate. The myoblast RNA device platform highlights a novel application of RNA controllers and provides insight for future RNA device design and characterization in mammalian synthetic biology.

## Materials and methods

### Plasmid construction

Transfection vectors were based on the donor plasmid pCS339 (pcDNA3.1). The EF1α-HRAS-sTRSVctrl cassette came from an EF1α-mCherry-sTRSVctrl; PGK-Clover bicistronic plasmid pCS4811. pCS4811 was modified with a truncated EF1α promoter from pCS3708 to form pCS4812. HRAS from MSCV H-Ras V12 IRES GFP was obtained as a gift from Scott Lowe (Addgene plasmid # 18780; http://n2t.net/addgene:18780; RRID:Addgene_18780) and was cloned into pCS4812 to form pCS4813 with the full EF1α-HRAS-sTRSVctrl cassette. This cassette was cloned into the MluI/XbaI sites in pCS339 to make pCS4814.

JAK1 from pDONR223-JAK1 was obtained as a gift from William Hahn and David Root (Addgene plasmid # 23932; http://n2t.net/addgene:23932; RRID:Addgene_23932) [36] and cloned into an EF1α-JAK1-sTRSVctrl cassette using the MluI/XhoI sites in pCS339 to make pCS4818. The sTRSV, TheoCAUAA, and TheoAGAAA switches were cloned into the SalI/AvrII sites in pCS4814 and pCS4818.

Primer sequences for cloning were obtained from the Protein and Nucleic Acid Facility at Stanford University. Enzymes for cloning, including restriction enzymes and Gibson assembly components, were obtained from New England BioLabs. All cloned constructs were sequence verified by QuintaraBio and purified with either Wizard Plus SV Minipreps DNA Purification System (Promega) or Plasmid Plus Midi Kit (Qiagen).

### Cell culture

Unless otherwise indicated, C2C12 cells were obtained from ATCC (CRL-1772) and maintained in growth medium (GM). GM consisted of Dulbecco's modified Eagle's medium (DMEM) with Glutamax (Thermo Fisher Scientific #10569044), 10% Fetal Bovine Serum (Thermo Fisher Scientific), and 1% penicillin/streptomycin (Thermo Fisher Scientific). Cells were induced to differentiate with differentiation medium (DM), consisting of DMEM without sodium pyruvate (Thermo Fisher Scientific #11965118), 2% Horse Serum (Gibco #26050070), and 1% penicillin/streptomycin. All cells were grown at 37°C, 5% $CO_2$, and 80% humidity in an incubator.

### Ligand preparation

Theophylline (catalog number T1633) was obtained from Sigma-Aldrich and was dissolved directly in either GM or DM to make a 1 mM stock solution. All ligand solutions were filter sterilized before use and freshly prepared before each experiment or stored at 4°C for less than one week.

### Flow cytometry

C2C12 cells were seeded in GM at 13,000 cells/well in 24-well plates. Twenty-four hours after seeding, cells were transfected using Lipofectamine 2000 (Thermo Fisher) according to the manufacturer's instructions using 500 ng DNA/well and 3 μL of Lipofectamine per well. Ligand was added 1 h before transfection. Forty-eight hours later, cells were trypsinized and resuspended in GM with no ligand. The final resuspended cells were assayed via flow cytometry on a MACSQuant flow cytometer using lasers with excitation wavelengths of 561 nm and 488 nm in channels Y2 (filter 615/20 nm) and B1 (filter 525/50 nm) respectively. Analysis of the data was performed in FlowJo, where cells are gated for viable cells and singlets.

Cellular fluorescence was determined by normalizing the fluorescence intensity of mCherry to the fluorescence intensity in the GFP channel to give the mean fluorescence ratio. This ratio

was then normalized further to that of the non-cleaving sTRSV ribozyme mutant sTRSVctrl to remove non-specific fluorescence effects and to standardize measurements across experiments. The resulting measurement was termed Relative Fluorescence Units (RFU).

## Myoblast differentiation

C2C12 cells were seeded in GM at 13,000 cells/well in 24-well plates. Twenty-four hours after seeding, cells were transfected using ViaFect Transfection Reagent (Promega) according to the manufacturer's instructions using 1000 ng DNA/well and 8 µL of ViaFect per well. Four hours after transfection, media was replaced with fresh GM containing 0 mM or 1 mM theophylline. Forty-eight hours after transfection, when cells had reached 90–100% confluency, the medium was replaced with fresh DM containing 0 mM or 1 mM theophylline. DM was replaced every 24 h until the indicated timepoint.

## Immunocytochemistry

For immunofluorescence analysis, C2C12 cultures were fixed at the indicated timepoints for 10 min in 4% paraformaldehyde at room temperature. Permeabilization and blocking of non-specific binding were done for 30 min with 2% bovine serum albumin (New England BioLabs) in PBS containing 0.1% Saponin (Sigma Aldrich). Samples were incubated for 60 min with primary antibody for Myosin Heavy Chain (Developmental Studies Hybridoma Bank) at 4 µg/mL diluted in the same solution. Samples were washed with the same solution three times before incubation with secondary antibody (ReadyProbes AlexaFluor 594) for 45 min at room temperature. Samples were washed with the same solution three times before incubation with DAPI at 10 µg/mL diluted in the same solution. Fluorescence was viewed with an AxioImager microscope courtesy of Stanford's Cell Sciences Imaging Facility.

## Fusion index

For fusion index determination, fluorescence images were collected and analyzed by counting the number of nuclei in myotubes (MyHC+ cells) and the total number of nuclei. The fusion index was calculated as the ratio of the number of myotube-associated nuclei to the total nuclei. Five randomly chosen fields from each well were counted at a magnification of 20X. Counting was done by hand using the cell counter program in FIJI.

## RNA extraction and RT-qPCR

C2C12 cells were seeded in GM at 13,000 cells/well in 24-well plates. Twenty-four hours after seeding, cells were transfected using ViaFect Transfection Reagent (Promega) according to the manufacturer's instructions using 1000 ng DNA/well and 8 µL of ViaFect per well. Four hours after transfection, media was replaced with fresh GM containing 0 mM or 1 mM theophylline. Forty-eight hours after transfection, when cells had reached 90–100% confluency, GM was replaced with fresh DM containing 0 mM or 1 mM theophylline. DM was replaced every 24 h until the indicated timepoint.

At the indicated timepoint, total RNA was extracted using the RNeasy Plus kit with QIAshredder (Qiagen) for homogenization following manufacturer's instructions. RT-qPCR analysis was performed in triplicate using Luna Universal One-Step RT-qPCR Kit (New England BioLabs) and a QuantStudio 3 (Applied Biosystems). Thirty nanograms of total RNA was added to each 20 µL reaction in a 96-well plate. mRNA levels were normalized to beta-actin and plotted as the fold change from the expression level of the indicated switch control (generally sTRSVctrl or sTRSV).

The following primer sequences used for RT-qPCR were obtained from the Protein and Nucleic Acid Facility at Stanford University.

ACTB1:

```
5'-GATCATTGCTCCTCCTGAGC-3' (fwd)
```

```
5'-ACATCTGCTGGAAGGTGGAC-3' (rev)
```

HRAS:

```
5'-CAAGACCCGGCAGGGAG-3' (fwd)
```

```
5'-ACACACTTGCAGCTCATGC-3' (rev)
```

JAK1:

```
5'-CCTAACTGTCCAGATGAGGTTTATCAAC-3' (fwd)
```

```
5'-AAAGTGCTTCAAATCCTTCAATAAGG-3' (rev)
```

MYOG:

```
5'-GGTCCCAACCCAGGAGATCAT-3' (fwd)
```

```
5'-TTCGTCTGGGAAGGCAACAG-3' (rev)
```

MyHC (Myh 1):

```
5'-ACCAAGGAGGAGGAACAGC-3' (fwd)
```

```
5'-GTTGAGTGAATGCCTGTTTGC-3' (rev)
```

## Supporting information

**S1 Fig. C2C12 myoblast viability in the presence of theophylline.** Dose response curve for C2C12 cell count showing relative cell count normalized to the cell count at 0 mM theophylline. Error bars indicate standard deviation of two biological replicates.
(TIF)

**S2 Fig. Comparison of sTRSV and sTRSVctrl constructs with WT. A.** Real-time RT-qPCR detection of Myogenin five days after transient transfection of HRAS and JAK1 constructs compared to wild-type myoblasts. **B.** Real-time RT-qPCR detection of MyHC five days after transient transfection of HRAS and JAK1 constructs compared to wild-type myoblasts. Error bars indicate standard deviation of four biological replicates.
(TIF)

**S3 Fig. Theophylline-responsive ribozyme switches control HRAS expression in a dose-dependent manner. A.** Dose response curve for TheoCAUAA ribozyme switch showing relative HRAS mRNA transcript levels normalized to transcript levels at 1 mM theophylline. **B.** Dose response curve for TheoAGAAA ribozyme switch showing relative HRAS mRNA transcript levels normalized to transcript levels at 1 mM theophylline. Error bars indicate standard deviation of three or more biological replicates.
(TIF)

**S1 File.**
(ZIP)

**S1 Dataset.**
(XLSX)

## Acknowledgments

We thank Drs. S.L. Delp, M.W. Covert, and E.A. Appel for valuable advice and input on the project. Microscopy for this project was performed on instruments in the Stanford Cell Sciences Imaging Facility. We thank Dr. R. Chen and C.S. Liou for advice on RT-qPCR procedures.

## Author Contributions

**Conceptualization:** Peter B. Dykstra, Thomas A. Rando, Christina D. Smolke.

**Formal analysis:** Peter B. Dykstra.

**Funding acquisition:** Peter B. Dykstra, Thomas A. Rando, Christina D. Smolke.

**Investigation:** Peter B. Dykstra.

**Methodology:** Peter B. Dykstra, Christina D. Smolke.

**Resources:** Christina D. Smolke.

**Supervision:** Thomas A. Rando, Christina D. Smolke.

**Writing – original draft:** Peter B. Dykstra.

**Writing – review & editing:** Thomas A. Rando, Christina D. Smolke.

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
