## [Decision Letter · Decision Letter 0]

12 Jul 2022

PONE-D-22-09291Modulating myoblast differentiation with RNA controllersPLOS ONE

Dear Dr. Smolke,

Thank you for submitting your manuscript to PLOS ONE. After careful consideration, we feel that it has merit but does not fully meet PLOS ONE’s publication criteria as it currently stands. Therefore, we invite you to submit a revised version of the manuscript that addresses the points raised during the review process.

Please implement all suggestions that require changes to text, as well as statistical analyses and interpretation of your findings. Please seriously consider adding the additional experiments suggested by one reviewer.

We look forward to receiving your revised manuscript.

Kind regards,

Thomas Preiss, PhD

Academic Editor

PLOS ONE

Journal Requirements:

Reviewers' comments:

Reviewer's Responses to Questions

**Comments to the Author**

1. Is the manuscript technically sound, and do the data support the conclusions?

Reviewer #1: Yes

Reviewer #2: Partly

2. Has the statistical analysis been performed appropriately and rigorously? 

Reviewer #1: Yes

Reviewer #2: I Don't Know

3. Have the authors made all data underlying the findings in their manuscript fully available?

Reviewer #1: Yes

Reviewer #2: Yes

4. Is the manuscript presented in an intelligible fashion and written in standard English?

Reviewer #1: Yes

Reviewer #2: Yes

5. Review Comments to the Author

Reviewer #1: Dykstra et al. characterise two perviously described theophylline-responsive ribozyme-based RNA switches in the mouse myoblast C2C12 cell line and use these constructs to control an anti-differentiation signal to allow the expansion of these cells in differentiation inducing media. They use fluorescence reporter to show that both of these theophylline-responsive ribozyme swithces are working in the C2C12 cell line as designed (mRNA is degraded/off without theophylline, while stabilised/on with theophylline). The authors then use these constructs to regulate the levels of HRAS or JAK1 mRNAs depending on the presence or absence of theophylline and show that the HRAS constarct can inhibit/slow down myoblast differentiation in a theophylline-dependent manner.

Overall, the study is scientifically solid and the conclusions are supported by the data presented in the manuscript.

Major comments:

- The authors use the HRAS-sTRSV or Jak1-sTRSV constructs to normalise all myoblast differentiation experiments (Fig 4 C-F and Fig 5), however, this might be misleading if the residual amount of HRAS or Jak1 signal that is still present in these constructs can already slow down differentiation. The fluorescence construct show ~10% signal in the sTRSV construct compared to the non-cleaving construct, while not likely, it is possible that already a low level of HRAS or Jak1 might have an effect on the differentiation of these cells. A simple control showing that the HRAS-sTRSV or Jak1-sTRSV constructs have no effect on the differentiation of these cells would further solidify the conclusions of the manuscript.

- It is a bit unclear what the major conceptual outcome of this study is - beyond that these ligand-responsive ribozyme switches work in these cell line? The idea that these constructs could be used in clinical applications is far-fetched, given the risks associated with transfecting cells with DNA constructs that contain a very strong oncogene. Would it be possible to transfect these constructs as RNA (probably already loaded with theophylline)? If yes, what would be the advanatge compared to just using HRAS or Jak1 mRNAs (without ligand-induced rybozyme switch) to delay differentiation? A short paragraph in the discussion would help the readers to better understand the significance of this study.

Minor comments:

- Fig 3B - The DAPI signal is barely visible on the Figure and it is very hard to notice that e.g. the nuclei number is increased in the hRAS sample.

- figure 4 legend : "Values are means normalized to either sTRSVctrl (A and D) or sTRSV (B, C, E, F);” is most likely a typo, A and B are normalized to sTRSVctrl and C-F are normalized to sTRSV

Reviewer #2: The Authors of the manuscript Modulating myoblast differentiation with RNA controllers (Smolke et al) have expended on their initial findings which since have been implemented in several cellular and non cellular models successfully. The present study use the same transfection system to inhibit c2c12 myoblasts into myotubes through response to exogenous theophylline supplement.

Major concerns:

While the general idea/goal of the study is clear and the proof of concept is somewhat obtained as per previous studies by the authors, what is unclear to this reviewer is the choice of the test genes for this specific cell line. C2c12 are robust and adjustable myoblasts used in many proof of concept studies. The myogenic pathway and other pathways dependent and interacting with this pathway are well known, as well as most genes playing part in these pathways. Hence, the logic of why use HRAS which is remotely associated with myogenic differentiation is unclear. Was the hypothesis that this "over arching" gene will have an effect on any differentiating cells. As it turned out to be, transfections with JAK1 which is associated with the myogenic pathway did not result in such a significant ability to switch on/off genes associated with myogenesis. It is of importance to do the same transfection in genes that are downstream and are directly related to the myogenic pathway (Pax7, MyoD etc).

An array of genes specifically related to the myogenic pathway in a stage specific manner can be tested, however, if the end point is fusion and formation of tubes, would the authors consider identifying the expression of genes directly associated with fusion (ie; myomaker etc). Also it is important to confirm gene expression with protein expression. Since only couple of genes were examined, this experiment should be added.

What is the reason of testing the ribozyme switch in PGK and EF1A first. The method was clearly repeated in the test genes (HRAS and JAK1). Instead a dose response to theophylline and cell viability experiments should be conducted. The authors indicate that there was some toxicity to theophylline, but except of indicating that this occurs above 1mM concentration and refer to other studies, they should show this clearly in the current study.

The authors indicate on 4 experimental groups (lines 196-200). Those grown in the presence or not of theophylline and then differentiated in the presence or not of theophylline. Which one of these groups are presented in figure 3?. Additionally, will cells grown in the presence of theophylline followed by removal of theophylline in a differentiation medium (transfected with either HRAS or JAK1), form tubes with a normal fusion index as WT?

The fact that JAK1-based controllers had negative results is an indication that the ribozyme switch may be effective in "master" controller genes such as HRAS acting in an "all or nothing" way such as in cancer and embryonic cells, but not in downstream genes which are more specific to pathways associated with tissue specific differentiation. This means that the method which is ingenious is not yet suitable to switch on/of genes related to tissue specific diseases, and in this case, such as muscle dystrophy.

Minor comments:

All figures:

Are the bars indicate SEM or SD?

Fig 2B concentrations of theophylline are in M and in 2C in uM. It will be useful to use similar terms as appear in all figures.

Figure 4E and F indicates significance of 0.01 in groups with either SEM or SD (???) which overlap. This needs to be retested. Was the T-test a paired T-Test and was a post-hoc used?.

6. PLOS authors have the option to publish the peer review history of their article (what does this mean?). If published, this will include your full peer review and any attached files.

Reviewer #1: No

Reviewer #2: No

---

## [Author Response · Author response to Decision Letter 0]

25 Aug 2022

Please see the provided response to reviewer comments document for detailed point-by-point responses to editor and reviewer comments. Thank you.

---

## [Decision Letter · Decision Letter 1]

13 Sep 2022

Modulating myoblast differentiation with RNA controllers

PONE-D-22-09291R1

Dear Dr. Smolke,

We’re pleased to inform you that your manuscript has been judged scientifically suitable for publication and will be formally accepted for publication once it meets all outstanding technical requirements.

Kind regards,

Thomas Preiss, PhD

Academic Editor

PLOS ONE

Additional Editor Comments (optional):

Reviewers' comments:

Reviewer's Responses to Questions

**Comments to the Author**

1. If the authors have adequately addressed your comments raised in a previous round of review and you feel that this manuscript is now acceptable for publication, you may indicate that here to bypass the “Comments to the Author” section, enter your conflict of interest statement in the “Confidential to Editor” section, and submit your "Accept" recommendation.

Reviewer #1: All comments have been addressed

Reviewer #2: (No Response)

2. Is the manuscript technically sound, and do the data support the conclusions?

Reviewer #1: Yes

Reviewer #2: Partly

3. Has the statistical analysis been performed appropriately and rigorously? 

Reviewer #1: Yes

Reviewer #2: I Don't Know

4. Have the authors made all data underlying the findings in their manuscript fully available?

Reviewer #1: Yes

Reviewer #2: Yes

5. Is the manuscript presented in an intelligible fashion and written in standard English?

Reviewer #1: Yes

Reviewer #2: Yes

6. Review Comments to the Author

Reviewer #1: (No Response)

Reviewer #2: The manuscript titled Modulating myoblast differentiation with RNA controllers by Dykstra et al had minor changes since the 1st review. Explanation has been included in the text replying to comments, however, no additional information has been provided assuming the authors chose not to add more experiments. saying that, an addition of three figures as supplements were also incorporated into the revised manuscript.

The firs added S1 should be removed as it based on 2 biological repeats and therefore it is not scientifically sound.

7. PLOS authors have the option to publish the peer review history of their article (what does this mean?). If published, this will include your full peer review and any attached files.

Reviewer #1: No

Reviewer #2: No

---

## [Editor Report · Acceptance letter]

19 Sep 2022

PONE-D-22-09291R1 

Modulating myoblast differentiation with RNA-based controllers 

Dear Dr. Smolke:

I'm pleased to inform you that your manuscript has been deemed suitable for publication in PLOS ONE. Congratulations! Your manuscript is now with our production department. 

Kind regards, 

on behalf of

Prof Thomas Preiss 

Academic Editor

PLOS ONE